# Combination Therapy after TACE for Hepatocellular Carcinoma with Macroscopic Vascular Invasion: Stereotactic Body Radiotherapy versus Sorafenib

**DOI:** 10.3390/cancers10120516

**Published:** 2018-12-14

**Authors:** Lujun Shen, Mian Xi, Lei Zhao, Xuhui Zhang, Xiuchen Wang, Zhimei Huang, Qifeng Chen, Tianqi Zhang, Jingxian Shen, Mengzhong Liu, Jinhua Huang

**Affiliations:** 1Department of Minimally Invasive Interventional Therapy, Sun Yat-sen University Cancer Center, Guangzhou 510060, China; shenlj@sysucc.org.cn (L.S.); wangxiuch@sysucc.org.cn (X.W.); huangzhm@sysucc.org.cn (Z.H.); chenqf25@sysucc.org.cn (Q.C.); zhangtq@sysucc.org.cn (T.Z.); 2State Key Laboratory of Oncology in South China, Collaborative Innovation Center of Cancer Medicine, Sun Yat-sen University, Guangzhou 510060, China; ximian@sysucc.org.cn (M.X.); zhaolei@sysucc.org.cn (L.Z.); zhangxuh@sysucc.org.cn (X.Z.); shenjx@sysucc.org.cn (J.S.); liumzh@sysucc.org.cn (M.L.); 3Department of Radiation Oncology, Sun Yat-sen University Cancer Center, Guangzhou 510060, China; 4Department of Radiology, Sun Yat-sen University Cancer Center, Guangzhou 510060, China

**Keywords:** hepatocellular carcinoma, macrovascular invasion, stereotactic body radiotherapy, sorafenib, transarterial chemoembolization

## Abstract

Stereotactic body radiotherapy (SBRT) has shown promising results in the control of macroscopic vascular invasion in patients with hepatocellular carcinoma (HCC); however, its efficacy in comparison to sorafenib when combined with transarterial chemoembolization (TACE) remains to be determined. Between 2009 and 2017, 77 HCC patients with macroscopic vascular invasion receiving TACE–SBRT or TACE–sorafenib combination therapies were enrolled. The best treatment responses, overall survival (OS), and progression-free survival (PFS) of the two treatment arms were compared. Of the patients enrolled, 26 patients (33.8%) received TACE–SBRT treatment, and 51 (66.2%) received TACE–sorafenib treatment. The patients in the TACE–SBRT group were more frequently classified as elder in age (*p* = 0.012), having recurrent disease (*p* = 0.026), and showing lower rates of multiple hepatic lesions (*p* = 0.005) than patients in TACE–sorafenib group. After propensity score matching (PSM), 26 pairs of well-matched HCC patients were selected; patients in the TACE–SBRT group showed better overall response rates in trend compared to those in the TACE–sorafenib group. The hazard ratio (HR) of OS to PFS for the TACE–SBRT approach and the TACE–sorafenib approach was 0.36 (95% CI, 0.17–0.75; *p* = 0.007) and 0.35 (95% CI, 0.20–0.62; *p* < 0.001), respectively. For HCC patients with macrovascular invasion, TACE plus SBRT could provide improved OS and PFS compared to TACE–sorafenib therapy.

## 1. Introduction

The macroscopic vascular invasion of hepatocellular carcinoma (HCC), including portal vein tumor thrombosis (PVTT) or inferior vena cava tumor thrombosis (IVCTT), corresponds to the Barcelona Clinic Liver Cancer (BCLC) stage C disease and remains a daunting challenge for treatment [1,2]. The current standard of care for HCC patients with PVTT/IVCTT is sorafenib, which is recommended as the only treatment option in EASL–EORTC practical guideline [3]. However, only modest clinical efficiency was observed when adopting sorafenib monotherapy for HCC patients with macrovascular invasion (MVI), with a median overall survival (OS) time of 8.1 months in the SHARP trial [4].

Transarterial chemoembolization (TACE) is the most popular palliative treatment for patients with unresectable HCC. In the past decade, the strategy of combining conventional TACE and sorafenib to treat HCC patients with macroscopic invasion had been proposed [5,6,7] and reported to prolong the time-to-progression (TTP) by 0.4 months and OS by three months when compared to sorafenib monotherapy [8]. The underlying rationale for developing this combination is that sorafenib can inhibit the tumor angiogenesis after TACE and lead to a synergistic therapeutic effect, which is supported by a preclinical study [9]. Although a survival benefit had been witnessed by adopting this combination, the survival of these HCC patients remained poor and a better combination was needed.

In recent years, the advances in radiotherapy techniques, including three-dimensional conformal radiotherapy (3D-CRT), have facilitated the safe and effective delivery of a high dose of radiation to the tumor while preserving the surrounding tissue [10]. A recent randomized controlled trial conducted in Korea involving HCC patients with macroscopic vascular invasion reported encouraging results where sequentially combined TACE and 3D-CRT was well tolerated, and the combined therapy improved progression-free survival (PFS), objective response rate, TTP, and OS when compared with those of sorafenib monotherapy [11]. Compared with 3D-CRT, stereotactic body radiotherapy (SBRT) is a more advanced form of radiation modality in which a high-radiation dose can be precisely concentrated on the tumor in a few fractions [10,12]. The sequential combination of TACE with SBRT has great potential to challenge the current golden combination of the TACE–sorafenib strategy in the management of HCC patients with PVTT/IVCTT [13].

Therefore, the goal of this study was to retrospectively compare the OS and PFS in advanced HCC patients with macroscopic vascular invasion who are undergoing combined TACE–SBRT treatment versus combined TACE–sorafenib treatment.

## 2. Results

### 2.1. Baseline Characteristics of the TACE–SBRT and TACE–Sorafenib Groups

Of the 77 patients enrolled, 26 (33.8%) were in the TACE–SBRT group and 51 (66.2%) were in the TACE–sorafenib group. Table 1 shows the baseline characteristics for the two groups. Before propensity score matching (PSM), patients who underwent TACE–SBRT treatment were more frequently classified as elder in age (*p* = 0.012), having recurrent disease (*p* = 0.026), and having single intrahepatic lesions (*p* = 0.005) than patients in the TACE–sorafenib group. There were no significant differences in the distribution of gender, Child–Pugh class, cause of disease, tumor size, alpha-fetoprotein (AFP) level, type of PVTT, and hepatic vein/inferior venae cava (HV/IVC) invasion between the two groups.

### 2.2. OS and PFS in the Whole Population

The mean follow-up time for the TACE–SBRT group and TACE–sorafenib group were 12.2 months (range, 1–60) and 10.4 months (range, 2–45), respectively. Among the 26 patients in the TACE–SBRT, seven were still alive, nine had died, and 10 were lost to follow-up by the end of this study. Among the 51 patients who received TACE–sorafenib, two were still alive, 39 had died, and 10 were lost to follow-up. The median OS time was 24.2 months for the TACE–SBRT group and 12.9 months for the TACE–sorafenib group. The one-, three-, and five-year OS rates were 81.8%, 49.4%, and 35.3% in the TACE–SBRT group, and 52.1%, 11.4%, and 0.0% in the TACE–sorafenib group, respectively. The TACE–SBRT group had significantly higher OS rates than the TACE–sorafenib group (*p* = 0.001; Figure 1A). In all patients, liver tumor size (≥5 cm versus <5 cm) and treatment (TACE–SBRT versus TACE–sorafenib) were significantly associated with the OS in the univariate analysis (Table 2). The multivariate analysis showed that treatment was the only independent predictor, while the liver tumor size was not an independent risk factor (*p* = 0.076; Table 2).

Progression of the disease was observed in 14 of the patients (53.8%) who underwent TACE–SBRT and 43 of the patients who underwent TACE–sorafenib (84.3%) during follow-up. The median PFS time were 10.1 months for TACE–SBRT groups and 3.6 months for the TACE–sorafenib group. The one-, three-, and five-year PFS rates were 44.7%, 26.2%, and 9.4% in the TACE–SBRT group, and 13.4%, 2.2%, and 0.0% in the TACE–sorafenib group, respectively. Patients in the TACE–SBRT group had significantly higher PFS rates compared with those in the TACE–sorafenib group (*p* < 0.001; Figure 1B). In all patients, liver tumor size (≥5 cm versus <5 cm) and treatment (TACE–SBRT versus TACE–sorafenib) were identified as independent prognostic factors for PFS (Table 3).

### 2.3. One-to-One Propensity Score Analysis

By utilizing one-to-one PSM, 26 patients from each group were selected, with well-matched confounding factors (Table 1). The response of the hepatic tumors and MVI in these two groups are shown in Table 4. The TACE–SBRT group had a higher overall response rate than the TACE–sorafenib group, but without a significant difference (42.3% versus 23.1%, *p* = 0.139).

The median OS and PFS time in the TACE–SBRT group were 24.2 and 10.0 months, and in the TACE–sorafenib group these were 8.4 and 3.5 months, respectively. The one-, three-, and five-year OS rates were 81.8%, 49.5%, and 35.3% in the TACE–SBRT group, and 46.9%, 7.8%, and 0.0% in the TACE–sorafenib group, respectively. The one-, three-, and five-year PFS rates were 44.7%, 26.2%, and 9.4% in the TACE–SBRT group, and 12.5%, 0.0%, and 0.0% in the TACE–sorafenib group, respectively. Patients in the TACE–SBRT group had significantly higher OS (*p* < 0.001) and PFS rates (*p* < 0.001) than patients in the TACE–sorafenib group (Figure 1C,D). The adjusted HR for the OS and PFS of TACE–SBRT to TACE–sorafenib was 0.26 (95% CI, 0.11–0.63; *p* = 0.003) and 0.31 (95% CI, 0.16–0.61; *p* = 0.001), respectively.

### 2.4. Follow-Up Treatments

The follow-up treatments in the two groups are listed in Table 5. There were no significant differences in the number of adopted treatment modalities between the two groups.

### 2.5. Toxicity of SBRT

No treatment-related grade 4 or 5 acute toxicity in liver function and bone marrow was observed within three months after SBRT for patients in the TACE–SBRT group (Table 6). Three patients (11.5%) had grade 3 leukocytopenia and three patients (11.5%) had grade 3 thrombocytopenia. One patient (3.8%) showed grade 3 elevation of bilirubin and one patient had grade 3 elevation of liver enzymes.

## 3. Discussion

In this study, we found that TACE sequentially combined with SBRT is a safe and effective combination strategy for HCC patients with MVI. The results of PSM suggested that TACE–SBRT combination therapy had superior therapeutic efficacy compared to TACE–sorafenib combination therapy in the treatment of HCC patients with macroscopic vascular invasion.

In Asian regions, TACE is considered the primary treatment for patients with unresectable HCC confined to the liver [14]. Two randomized trials comparing TACE to conservative treatment in HCC patients with PVTT consistently demonstrated favorable OS in the TACE arm [15,16]. Follow-up efforts had been made to improve the embolization efficacy for unresectable HCC through using drug-eluting beads and yttrium-90 radio-embolization; however, the achieved survival benefits were limited, as reported by a recently published network meta-analysis [17].

Sorafenib has been considered the first-line treatment for HCC patients with MVI in most guidelines [3,18,19]. Past studies suggest that TACE–sorafenib therapy could be a promising treatment combination for advanced stage HCC [20]. In an Asia-Pacific trial [21], the median OS time for sorafenib monotherapy was 6.5 months, while the median OS for TACE–sorafenib combination therapy was reported to range from to 8.9 to 11.0 months [8,22]. However, until now, neither TACE nor sorafenib had been reported to provide a durable treatment response for tumor thrombus of HCC [23]. Macrovascular invasion of HCC could not only promote intrahepatic dissemination of tumor cells [24], but could also contribute to portal hypertension, leading to the deterioration of liver function and subsequent complications including ascites, variceal hemorrhage, and thrombocytopenia induced by splenomegaly [25]; these complications could impede further treatment. Therefore, a good local control of the tumor thrombus is important in the management of advanced HCC with macrovascular invasion. In our study, a better overall response rate as well as significantly favorable survival were observed in patients in the TACE–SBRT group compared with those in TACE–sorafenib group. The survival benefits of the TACE–SBRT combination therapy could be explained by the fact that SBRT is an advanced radiation modality that can concentrate a high radiation dose precisely on the tumor in a few fractions, provide a high local control rate (>80%) for tumor thrombosis, while TACE provides good control of tumors outside the radiation field as a complement. The combination of TACE and SBRT enabled a balance in the control of HCC and the maintenance of liver function, thereby allowing additional treatments [11,26], including subsequent curative surgical resection, as shown in our study.

In our study, the OS and PFS for the TACE–SBRT group were 24.2 and 10.1 months, respectively, which were higher than 12.7 and 7.2 months for patients treated by TACE-3D-CRT in the recently published study by Yoon et al. [11]. This difference may be due to the fact that in our study, patients with lymph node metastasis and distant metastasis were excluded; moreover, the median tumor size in the TACE–SBRT group in our study was 7.7 cm, which was lower than the 9.8 cm in their study. The encouraging results in our study imply that HCC patients with a small tumor size and macroscopic vascular invasion and without extrahepatic lesions were the ideal population for receiving TACE–SBRT combination therapy. On the other side, our results might also suggest that TACE–SBRT can provide better local control than TACE-3D-CRT, as the overall response rate of 42.3% in the TACE–SBRT group in our study was higher than the response rate of 33.3% in the TACE-3D-CRT group reported by Yoon et al.

Although TACE–SBRT combination therapy had promising therapeutic efficacy in HCC patients with MVI, the median survival time in this group is still shorter. Further enhancing the treatment efficacy may require additional combinations. TACE-combined ablative therapies like radiofrequency ablation have been widely recognized for their efficacy in controlling intrahepatic lesions of HCC [2,27,28]. The combination of TACE, ablative therapies, and SBRT has the potential to achieve better local control than TACE–SBRT alone in a minimally invasive way. However, the ways to combine these three modalities require further research. In the area of immunotherapy, Kim et al. reported preclinical data on the improved anti-tumor effect of checkpoint inhibitors after radiation in HCC [29]. Hence, combining PD1/PD-L1 antibodies following TACE–SBRT therapy could also be a direction for research.

For the HCC patients who underwent SBRT, image-guided radiation therapy (IGRT) is mandatory and fundamental to ensure precision. The key to safe SBRT is to match the target in the planning CT to the target itself on the cone-beam CT before each fraction [30]. Since the liver lesions are usually invisible on the cone-beam CT without contrast injection, the use of implanted fiducials in and/or around the tumor as surrogates is a reliable way to assess respiratory motion during treatment, especially for robotic-based liver SBRT [31]. However, only a minority of patients choose this manner in clinical practice because internal fiducials are invasive to insert and have a relatively higher cost. Therefore, for patients who received TACE as initial treatment before radiotherapy, we usually used embolization material as a surrogate for tumor position at our institution.

This study had several limitations. First, it is a retrospective study. Second, the sample size of patients enrolled in each group was relatively small, which impeded further stratified analysis. Third, all TACE procedures were performed in our cancer center, which may cause bias as TACE is operator dependent. For these reasons, a multi-institutional clinical trial is needed in the future.

## 4. Materials and Methods

### 4.1. Study Design

From January 2009 to January 2017, a consecutive series of 746 HCC patients with macroscopic vascular invasion were retrospectively reviewed. This study was approved by the SYSUCC Hospital Ethics Committee (No. B2018-156-01), which waived the need for written informed consent because of the retrospective nature of the study. The inclusion criteria were: (a) initially diagnosed HCC with macroscopic vascular invasion or recurrent HCC after surgical resection with macroscopic vascular invasion; (b) received TACE as initial treatment and achieved technical success; (c) potential candidates for SBRT of vascular tumor thrombi evaluated by two radiation oncologists with more than 10 years’ experience; (d) received SBRT of vascular invasion within 3 months after first TACE or started sorafenib treatment within 1 week after first TACE as combination treatment; and (e) Child–Pugh A or B. The exclusion criteria were: (a) presence of extrahepatic metastasis detected on abdominal CT/MRI scan or chest CT scan/digital chest radiograph; (b) recurrent tumors with no intrahepatic lesions; (c) received other treatment including ablative therapies, surgical resection, transplantation within the first 6 months after initial TACE; and (d) other malignancies. A total of 77 patients were enrolled in this study with 26 in the TACE–SBRT group and 51 in the TACE–sorafenib group (Figure 2).

In this study, each case was fully evaluated by a multidisciplinary team before the initiating treatment, including interventional radiologists, oncologists, pathologists, radiologists, and radiation oncologists. Each patient was informed of the advantages and disadvantages of the two treatment plans before initial TACE, including costs, treatment-related morbidities, and treatment outcomes. The decision regarding the treatment plan was made jointly by the patient and doctors.

### 4.2. Transarterial Chemoembolization

For the TACE procedure, based on the tumor size, location, number, and vascular supply, a super-selective microcatheter was inserted into the supplying artery of the tumor. Then a combination of lipiodol (5–15 mL), lobaplatin (30–50 mg) and pirarubicin (30–50 mg) was introduced into the tumor. Technical success was defined as full embolization of the tumor feeding artery and with no tumor staining observed by angiogram at the end of procedure. Transarterial chemoembolization was repeated every 4–5 weeks thereafter, and was discontinued if the patients could not tolerate additional procedures due to adverse effects or when they refused further treatment. For the initial 6 months, the mean repeats of TACE for patients in the TACE–SBRT group was 1.50 (range, 1–2) before SBRT and 0.3 (range, 0–3) after SBRT (Appendix A); the mean repeats of TACE for patients in the TACE–sorafenib group was 2.1 (range, 1–6).

### 4.3. Stereotactic Body Radiotherapy

In the TACE–SBRT group, SBRT was scheduled after one to two administrations of TACE and completed within 3 months after the initial TACE. All patients underwent four-dimensional computed tomography (4DCT) scans at 2.5- or 3-mm slice thickness during free breathing. The tumor thrombosis visualized on the CT/MRI images and the contiguous HCC were defined as gross tumor volume (GTV) for all patients; the intrahepatic metastases of HCC were not included in GTV. The internal target volume (ITV) was defined as the combined volume of GTVs in the multiple 4DCT phases. To take interfractional motion variability and daily setup errors into account, the planning target volume (PTV) was acquired by adding a 6-mm margin to the ITV.

Volumetric modulated arc therapy (VMAT) techniques, which typically was planned for a 6 MV photons Elekta accelerator (Versa HD^TM^, Stockholm, Sweden), was utilized to perform SBRT. Planning of VMAT was conducted utilizing Monaco TPS (CMS, Elekta, Stockholm, Sweden) and consisted of a single arc [32].

The median prescription dose was 36 Gy (range, 30–42 Gy) to PTV in six fractions, 3 days per week. The total radiation dose was determined on the basis of tumor size, tumor location, and patients’ Child–Pugh Class. The dose constraints for organs at risk were defined as follows: mean dose of normal liver <18 Gy, mean dose of kidneys <15 Gy, maximum dose to spinal cord <27 Gy, and maximum dose to >0.5 cc of esophagus, stomach, or small intestine <30 Gy. Daily image guidance was performed using cone-beam CT for alignment of liver anatomy prior to each fraction.

### 4.4. Sorafenib Administration

For patients in the TACE–sorafenib group, sorafenib was continuously administered orally at a dosage of 400 mg twice a day (800 mg/day). Dosage reductions and termination of treatment were permitted if unacceptable treatment-related toxicity occurred or disease progressed.

### 4.5. Follow-Up and Endpoints

Patients in both groups were followed up and evaluated for response every 4–5 weeks during sessions of treatment and then every 3 months until death through contrast-enhanced CT or MR scans. The primary endpoint was overall survival (OS), which was calculated from the date of the initial treatment to death by any causes. The secondary endpoint was progression-free survival (PFS), which was calculated from the date of the initial treatment to disease progression or death.

Tumor response assessments were made by two independent radiologists using the modified Response Evaluation Criteria in Solid Tumors (mRECIST), who were blinded to each other’s opinion [33]. According to mRECIST criteria, tumor thrombosis (TT) should be considered a nontarget lesion. Complete response (CR) was defined as the disappearance of TT; incomplete response/stable disease was the persistence of TT; progressive disease was the unequivocal progression of TT. The best treatment response at 24 weeks after initial TACE was assessed for each patient in the two treatment groups. The response rate (RR) was defined as the proportion of patients with complete response (CR) or partial response (PR) in each treatment group.

### 4.6. Statistical Analysis

Categorical variables between treatment groups were compared using Pearson χ^2^ test or Fisher’s exact test. Ordinal variables between groups were compared using Wilcoxon rank sum test. OS and PFS were estimated by Kaplan–Meier method and the differences between the groups were compared using the log-rank test. Multivariate Cox regression model in backward conditioned method was utilized to determine the independent prognostic factors for survival.

Propensity score matching analysis was performed to reduce the potential bias in patient selection. The selected variables entered into the propensity model included age, gender, disease onset, Child–Pugh Class, AFP level, liver tumor size, number of tumors, type of PVTT and HV/IVC invasion. One-to-one matching between the groups was accomplished using the nearest-neighbor method, with the caliper of 0.25. Following propensity score matching, the adjusted comparisons were based on data from 26 patients in each treatment arm. Statistical analyses were performed using SPSS 21.0 or R 3.3.2 (The R Foundation for Statistical Computing, 2018). The value *p* < 0.05 was considered statistically significant.

The authenticity of this article has been validated by uploading the key raw data onto the Research Data Deposit public platform (www.researchdata.org.cn), with the approval RDD number as RDDA2018000943.

## 5. Conclusions

This retrospective study provided encouraging evidence that the combination therapy of TACE–SBRT can significantly enhance the control of tumor thrombus and prolong the OS and PFS in HCC patients with macroscopic vascular invasion when compared to TACE–sorafenib therapy. Further randomized control trials are needed to confirm our findings.

## Figures and Tables

**Figure 1 cancers-10-00516-f001:**
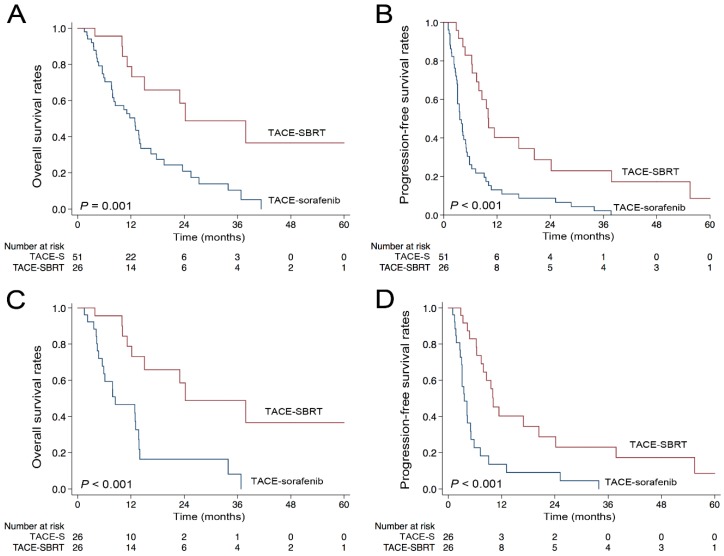
Kaplan–Meier curves of hepatocellular carcinoma (HCC) patients by different combination therapies in (**A**) overall survival (OS) before matching; (**B**) progression-free survival (PFS) before matching; (**C**) OS after matching; (**D**) PFS after matching. Note: TACE-S, TACE combined sorafenib therapy; TACE–SBRT, TACE sequentially combined SBRT therapy.

**Figure 2 cancers-10-00516-f002:**
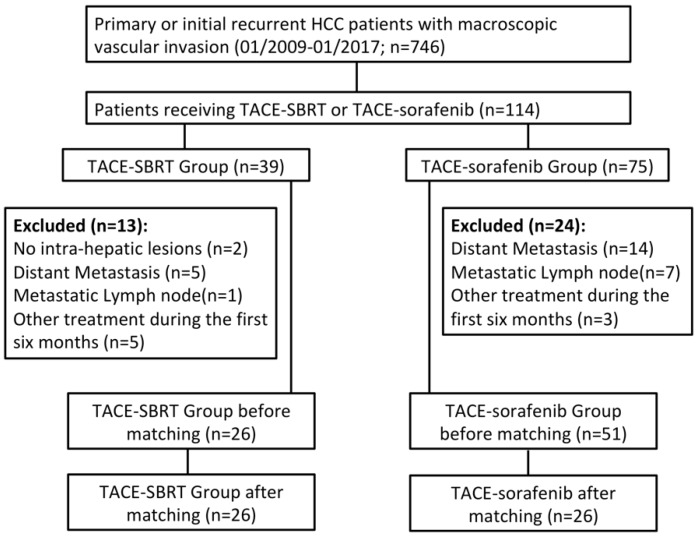
Flowchart of study design. A consecutive series of 746 HCC patients with macroscopic vascular invasion were reviewed and 77 were finally included based on the inclusion and exclusion criteria. Twenty-six patients were selected in each arm after 1:1 PSM.

**Table 1 cancers-10-00516-t001:** Baseline patient characteristics.

Variable	Before PSM			After PSM		
	TACE–sorafenib (*n* = 51)	TACE–SBRT (*n* = 26)	*p*	TACE–sorafenib (*n* = 26)	TACE–SBRT (*n* = 26)	*p*
Gender			0.657 *			1.000 *
Male	46 (90.2)	25 (96.2)		24 (92.3)	25 (96.2)	
Female	5 (9.8)	1 (3.8)		2 (7.7)	1 (3.8)	
Age (years)			0.012			0.532
<50	27 (52.9)	6 (23.1)		8 (26.7)	6 (23.1)	
≥50	24 (47.1)	20 (76.9)		22 (73.3)	20 (76.9)	
Child–Pugh Class			1.000			0.610 *
A	48 (94.1)	25 (96.2)		23 (88.5)	25 (96.2)	
B	3 (5.9)	1 (3.8)		3 (11.5)	1 (3.8)	
Cause of disease			0.338 *			1.000 *
HBV infection	51 (100.0)	25 (96.2)		26 (100.0)	25 (96.2)	
HCV infection	0 (0.0)	1 (3.8)		0 (0.0)	1 (3.8)	
Disease Onset			0.026 *			0.159
Initial	48 (94.1)	19 (73.1)		23 (88.5)	19 (73.1)	
Recurrent	3 (5.9)	7 (26.9)		3 (11.5)	7 (26.9)	
Tumor Size			0.344 *			0.465 *
<5 cm	7 (13.7)	6 (23.1)		3 (11.5)	6 (23.1)	
≥5 cm	44 (86.3)	20 (76.9)		23 (88.5)	20 (76.9)	
Number of Tumors			0.005			0.548
Single	20 (39.2)	19 (73.1)		17 (65.4)	19 (73.1)	
Multiple	31 (60.8)	7 (26.9)		9 (34.6)	7 (26.9)	
AFP level (ng/mL)			0.164			0.405
≤400	19 (37.3)	14 (53.8)		11 (42.3)	14 (53.8)	
>400	7 (62.7)	12 (46.2)		15 (57.7)	12 (46.2)	
Type of PVTT			0.449 **			0.864 **
Absent	3 (5.9)	2 (7.7)		0 (0.0)	2 (7.7)	
Type I/II	34 (66.7)	14 (53.8)		18 (69.2)	14 (53.8)	
Type III/IV	14 (27.5)	10 (38.5)		8 (30.8)	10 (38.5)	
HV or IVC invasion			0.526 *			0.419 *
Absent	44 (86.3)	21 (80.8)		24 (92.3)	21 (80.8)	
Present	7 (13.7)	5 (19.2)		2 (7.7)	5 (19.2)	

* Fisher’s exact test; ** Wilcoxon rank sum was used. Abbreviations: AFP, alpha-fetoprotein; HBV, hepatitis B virus; HCV, hepatitis C virus; PSM, propensity score matching; PVTT, portal vein tumor thrombosis; SBRT, Stereotactic body radiotherapy; TACE, transarterial chemoembolization.

**Table 2 cancers-10-00516-t002:** Univariate and multivariate analysis of OS in the enrolled cohort.

Variable	No. of Cases	Univariate Analysis		Multivariate Analysis	
		HR (95% CI)	*p* Value	HR (95% CI)	*p* Value
Gender (Female vs. Male)	6/71	0.92 (0.36–2.35)	0.866	–	–
Age (≥50 vs. <50)	44/33	0.64 (0.36–1.15)	0.136	–	–
Child–Pugh Class (B vs. A)	4/73	2.56 (0.77–8.49)	0.125	–	–
Disease Onset (Recurrent vs. Initial)	10/67	0.42 (0.13–1.36)	0.421	–	–
Liver Tumor Size (≥5 cm vs. <5 cm)	64/13	3.37 (1.20–9.43)	0.021	2.57 (0.91–7.29)	0.076
Number of Tumors (Multiple vs. Single)	39/38	0.99 (0.56–1.75)	0.979	–	–
AFP Level (>400 ng/mL vs. ≤400 ng/mL)	44/33	1.62 (0.89–2.95)	0.114	–	–
Type of PVTT (Type III/IV vs. Type I/II/Absent)	24/53	1.03 (0.56–1.91)	0.917	-	-
HV or IVC invasion (Present vs. Absent)	12/65	0.70 (0.30–1.66)	0.701	-	-
Treatment (TACE–SBRT vs. TACE–sorafenib)	26/51	0.30 (0.14–0.64)	0.002	0.36 (0.17–0.75)	0.007

Abbreviations: HR, hazard ratio.

**Table 3 cancers-10-00516-t003:** Univariate and multivariate analysis of PFS in the enrolled cohort.

Variable	No. of Cases	Univariate Analysis		Multivariate Analysis	
		HR (95% CI)	*p* Value	HR (95% CI)	*p* Value
Gender (Female vs. <Male)	6/71	0.97 (0.39–2.44)	0.948	–	–
Age (≥50 vs. <50)	44/33	0.74 (0.45–1.20)	0.220	–	–
Child–Pugh Class (B vs. A)	4/73	1.90 (0.68–5.30)	0.223	–	–
Disease Onset (Recurrent vs. Initial)	10/67	0.62 (0.28–1.37)	0.234	–	–
Liver Tumor Size (≥5 cm vs. <5 cm)	64/13	2.55 (1.21–5.39)	0.014	2.19 (1.03–4.64)	0.041
Number of Tumors (Multiple vs. Single)	39/38	1.43 (0.88–2.32)	0.152	–	–
AFP Level (>400 ng/mL vs. ≤400 ng/mL)	44/33	1.50 (0.88–2.38)	0.141	–	–
Type of PVTT (Type III/IV vs. Type I/II/Absent)	24/53	0.75 (0.44–1.29)	0.301	–	–
HV or IVC invasion (Present vs. Absent)	12/65	0.88 (0.45–1.72)	0.702	–	–
Treatment (TACE–SBRT vs. TACE–sorafenib)	26/51	0.32 (0.18–0.58)	<0.001	0.35 (0.20–0.62)	<0.001

**Table 4 cancers-10-00516-t004:** Best treatment response at 24 weeks since first TACE in the two treatment groups after propensity score matching.

Treatment	CR (*n*)	PR (*n*)	SD (*n*)	PD (*n*)	NA	RR (%)	*p* *
**Overall Response**							0.139
TACE–SBRT	1	10	5	7	3	42.3	
TACE–sorafenib	1	5	6	8	5	23.1	
**Response of Hepatic Lesions**							0.575
TACE–SBRT	4	8	5	6	3	46.2	
TACE–sorafenib	2	8	3	7	5	38.5	
**Response of Macrovascular Invasion**							/
TACE–SBRT	3	18 ^a^		2	3	/	
TACE–sorafenib	1	15 ^a^		4	5	/	

Note: ^a^ Sum of number of cases with PR or SD. * *p* value was calculated by comparing the RR of two groups using Chi-square test. CR, complete response; PR, partial response; SD, stable disease; PD, progression disease; NA, not available; RR, response rate.

**Table 5 cancers-10-00516-t005:** Follow-up treatment after initial combination therapy and PSM in the two treatment groups.

Categories	TACE–sorafenib (*n*, %)	TACE–SBRT (*n*, %)	*p*
Surgical resection	0 (0.0)	1 (3.8)	1.000 *
Ablative therapies	9 (34.6)	6 (23.1)	0.358
Iodine 125 seed implantation	1 (3.8)	0 (0.0)	1.000 *
Cytokine-induced killer cells infusion	0 (0.0)	1 (3.8)	1.000 *
Intra-arterial infusion	0 (0.0)	1 (3.8)	1.000 *

* Fisher’s exact test.

**Table 6 cancers-10-00516-t006:** Acute adverse effects on liver function and bone marrow in the TACE–SBRT group after SBRT.

Category and Grade *	Number (%)
Liver enzyme	
2	1 (3.8)
3	1 (3.8)
Bilirubin	
2	2 (7.7)
3	1 (3.8)
Albumin	
2	1 (3.8)
3	0 (0.0)
Leukocyte	
2	3 (11.5)
3	3 (11.5)
Platelets	
2	4 (15.4)
3	3 (11.5)

* The grading of adverse effects was based on the Common Terminology Criteria for Adverse Events (CTCAE) V4.02.

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
