# Peer review of "Combination Therapy after TACE for Hepatocellular Carcinoma with Macroscopic Vascular Invasion: Stereotactic Body Radiotherapy versus Sorafenib"

_cancers, 2018, doi:10.3390/cancers10120516_

Round 1

Reviewer 1 Report

Dear Ladies and Gentlemen

- please discuss the role of STATE, START and ART-scores to prove eligibility for TACE?

- please discuss potential differences in outcome using DEB TACE with doxorubicin versus cTACE with lobaplatin and pirarubcin

- please discuss the necessity to implant gold-marker for the SBRT-guidance?

Best regards

Author Response

Point 1. please discuss the role of STATE, START and ART-scores to prove eligibility for TACE?

Response 1: Many thanks for this suggestion. The STATE score and the use of ART score (START strategy) are designed to guide eligibility for TACE in patients with BCLC stage A or B HCC [1-2]. In our study, the HCC patients with macrovascular invasion included were classified as BCLC stage C. Currently, there is no published paper in validating the STATE, START and ART-scores in guiding TACE for BCLC stage C patients. It would be interesting and probably we can try to validate whether STATE, START and ART-scores could be utilized in doing so in the future. On the other hand, discussion it in the current manuscript is beyond the scope of our study.

1.Hucke F, Pinter M, Graziadei I, Bota S, Vogel W, Muller C, et al. How to STATE suitability and START transarterial chemoembolization in patients with intermediate stage hepatocellular carcinoma. J Hepatol. 2014;61(6):1287–96.

2.Sieghart W, Hucke F, Pinter M, Graziadei I, Vogel W, Muller C, et al. The ART of decision making: retreatment with transarterial chemoembolization in patients with hepatocellular carcinoma. Hepatology. 2013;57(6):2261–73.

Point 2. please discuss potential differences in outcome using DEB TACE with doxorubicin versus cTACE with lobaplatin and pirarubcin

Response 2: Many thanks for this kind suggestion. Four trials in the past had been published in comparing the efficacy of DEB-TACE versus cTACE in HCC in the past years, while no significant difference in local control as well as survival had been observed [1-4]. On the other side, it is reported that the post-procedural pain may be more frequent and severe after cTACE [1].

As the second reviewer also proposed to include the discussion of radioembolization in the manuscript, we added a paragraph including the discussion of both DEB-TACE and Radioembolization in the“Discussion Section”as bellowed:

In Asian regions, TACE is considered the primary treatment for patients with unresectable HCC confined in the liver. Two randomized trials comparing TACE to conservative treatment in HCC patients with PVTT had consistently demonstrated favorable OS in the TACE arm. Following efforts had been made to improve the efficacy of embolization for unresectable HCC through using drug-eluting beads and yttrium-90 radio-embolization, while the survival benefits were limited as reported by a recent published network meta-analysis.

References:

1.Golfieri R, Giampalma E, Renzulli M, Cioni R, Bargellini I, Bartolozzi C, et al. Randomised controlled trial of doxorubicin-eluting beads vs conventional chemoembolisation for hepatocellular carcinoma. British journal of cancer. 2014; 111(2):255-64.

2.Lammer J, Malagari K, Vogl T, Pilleul F, Denys A, Watkinson A, et al. Prospective randomized study of doxorubicin-eluting-bead embolization in the treatment of hepatocellular carcinoma: results of the PRECISION V study. Cardiovasc Intervent Radiol. 2010; 33(1):41-52.

3. Sacco R, Bargellini I, Bertini M, Bozzi E, Romano A, Petruzzi P, et al. Conventional versus doxorubicin-eluting bead transarterial chemoembolization for hepatocellular carcinoma. Journal of vascular and interventional radiology: JVIR. 2011; 22(11):1545-52.

4. van Malenstein H, Maleux G, Vandecaveye V, Heye S, Laleman W, van Pelt J, et al. A randomized phase II study of drug-eluting beads versus transarterial chemoembolization for unresectable hepatocellular carcinoma. Onkologie. 2011; 34(7):368-76.

Point 3. please discuss the necessity to implant gold-marker for the SBRT-guidance?

Response 3: For HCC patients who underwent SBRT, image-guided radiation therapy (IGRT) is mandatory and fundamental to ensure precision [1]. The key to safe SBRT is to match on the target on the planning CT to the target itself on the cone-beam CT before each fraction. Since the liver lesions are usually invisible on the cone-beam CT without contrast injection, the use of implanted fiducials in and/or around the tumor as surrogates is a reliable way to assess respiratory motion during treatment, especially for robotic-based liver SBRT [2]. However, only a minority of patients choose this manner in clinical practice because internal fiducials are invasive to insert and have a relative high cost. Therefore, for patients who received TACE as initial treatment before radiotherapy, we usually used embolization material as a surrogate for tumor position at our institution.

We added a paragraph discussing this issue in the “Discussion Section” as bellowed:

For HCC patients who underwent SBRT, image-guided radiation therapy (IGRT) is mandatory and fundamental to ensure precision. The key to safe SBRT is to match on the target on the planning CT to the target itself on the cone-beam CT before each fraction. Since the liver lesions are usually invisible on the cone-beam CT without contrast injection, the use of implanted fiducials in and/or around the tumor as surrogates is a reliable way to assess respiratory motion during treatment, especially for robotic-based liver SBRT. However, only a minority of patients choose this manner in clinical practice because internal fiducials are invasive to insert and have a relative high cost. Therefore, for patients who received TACE as initial treatment before radiotherapy, we usually used embolization material as a surrogate for tumor position at our institution.

1.    Riou O, Llacer Moscardo C, Fenoglietto P, Deshayes E, Tetreau R, Molinier J, Lenglet A, Assenat E, Ychou M, Guiu B, Aillères N, Bedos L, Azria D. SBRT planning for liver metastases: A focus on immobilization, motion management and planning imaging techniques. Rep Pract Oncol Radiother. 2017;22:103-110. doi: 10.1016/j.rpor.2017.02.006.

2.    Sterzing F, Brunner TB, Ernst I, Baus WW, Greve B, Herfarth K, Guckenberger M. Stereotactic body radiotherapy for liver tumors: principles and practical guidelines of the DEGRO Working Group on Stereotactic Radiotherapy. Strahlenther Onkol. 2014;190:872-81. doi: 10.1007/s00066-014-0714-1.

Reviewer 2 Report

The authors present a retrospective analysis of patients treated at 1 center for HCC with macrovascular invasion using 2 different combination : either TACE-sorafenib (considered standard) or TACE-SBRT. Propensity score matching was done. Number are relatively low but this is a subgroup with high unmet need. The results suggest superiority of TACE-SBRT over TACE-sorafenib.

Some comments:

Major comments:

1- My main concern regards the description of SBRT treatment, and more precisely the definition of GTV: "The tumor thrombosis visualized on the CT/MRI images and/or the primary liver lesions were defined as gross tumor volume (GTV), as determined by the radiation oncologists". I think this should be clarified. The authors should at least described how many patients had treatment to the thrombosis, how many had treatment to the liver lesions. It should be clarified whether the SBRT aimed to treat all viable lesions, or only a portion. I would have think that this strategy would be interesting mainly if the SBRT treated at least all the MVI, while the liver lesions might not be treated if TACE has resulted in response? But the wording suggest that for some patients tumor thrombosis was not in the GTV. I would suggest that such patients be excluded from analysis.

2- The authors consider TACE-sorafenib as standard. While I understand it is common practice in Asia, most guidelines still consider sorafenib as the only treatment option for patients with MVI. This should be further emphasize in the introduction and discussion.

3- I'm not sure that radiological endpoints (PFS, response rates) are easy to assess when dealing with MVI and SBRT treatment. The authors state that they assessed response of MVI separately by mRECIST. Most of their patients had type I to II PVT, which would be most of the time not target lesion by mRECIST (less than 1cm, difficult to ensure arterial enhancement). Moreover, in my experience, evaluation by mRECIST is particularly difficult following SBRT, with frequent inflammation mimicking arterial enhancement of the lesion. Comparison of PFS between a SBRT and a non-SBRT arm are particularly prone to evaluation bias. I would suggest to amend the text to recognize difficulty of evaluation as limitations, and please clarify if MVI was really evaluated by mRECIST as target lesions?

4- As the treatment was performed in one single center, there is a risk for selection bias. This has been mitigated by propensity score matching. However, could the authors better explain how the treatment modalities were proposed to patients?

Minor comments:

1- Please clarify the presence of metastasis as an exclusion criteria

2- Please add in the introduction and/or discussion the median OS obtained for patients with MVI for sorafenib in the SHARP (about 8 months) and if available in the AP trial

3- I was not able to review the suppl material table; however the figures are give in the text, so I'm not sure it adds any important data.

4- Radioembolization is an alternative for such patients, and could be discussed.

Author Response

Response to Reviewer 2 Comments

The authors present a retrospective analysis of patients treated at 1 center for HCC with macrovascular invasion using 2 different combination : either TACE-sorafenib (considered standard) or TACE-SBRT. Propensity score matching was done. Number are relatively low but this is a subgroup with high unmet need. The results suggest superiority of TACE-SBRT over TACE-sorafenib.

Reviewer #2

Some comments:

Major comments:

Point 1. My main concern regards the description of SBRT treatment, and more precisely the definition of GTV: "The tumor thrombosis visualized on the CT/MRI images and/or the primary liver lesions were defined as gross tumor volume (GTV), as determined by the radiation oncologists". I think this should be clarified. The authors should at least described how many patients had treatment to the thrombosis, how many had treatment to the liver lesions. It should be clarified whether the SBRT aimed to treat all viable lesions, or only a portion. I would have think that this strategy would be interesting mainly if the SBRT treated at least all the MVI, while the liver lesions might not be treated if TACE has resulted in response? But the wording suggest that for some patients tumor thrombosis was not in the GTV. I would suggest that such patients be excluded from analysis.

Response 1: We are sorry for the misunderstanding caused by the wording in the above sentence, which should be corrected as “The tumor thrombosis visualized on the CT/MRI images and the contiguous HCC were defined as gross tumor volume (GTV) for all patients; the intrahepatic metastases of HCC were not included in GTV”. Actually, all the patients in the TACE-SBRT group had SBRT treatment to tumor thrombosis. Internal target volume (ITV) was defined as the combined volume of GTVs in the multiple 4DCT phases. To take interfractional motion variability and daily setup errors into account, the planning target volume (PTV) was acquired by adding a 6-mm margin to the ITV.

We have corrected the wording in the “Methods Section” of the revised manuscript as bellowed:

The tumor thrombosis visualized on the CT/MRI images and the contiguous HCC were defined as gross tumor volume (GTV) for all patients; the intrahepatic metastases of HCC were not included in GTV. Internal target volume (ITV) was defined as the combined volume of GTVs in the multiple 4DCT phases. To take interfractional motion variability and daily setup errors into account, the planning target volume (PTV) was acquired by adding a 6-mm margin to the ITV.

We hope the revised description of GTV can be more easily understood.

Point 2. The authors consider TACE-sorafenib as standard. While I understand it is common practice in Asia, most guidelines still consider sorafenib as the only treatment option for patients with MVI. This should be further emphasize in the introduction and discussion.

Response 2: Many thanks for this suggestion. So far, sorafenib is recommended as the first-line treatment for advanced stage hepatocellular carcinoma recommended in nearly all the guidelines for HCC (including Chinese guidelines for primary liver cancer 2017, Consensus Guideline by Japan Society of Hepatology, NCCN guidelines 2018 and EASL-EORTC practical guidelines), and it is the only treatment option recommended by the EASL-EORTC practical guidelines. In the revised manuscript, we emphasize the importance of sorafenib in the “Introduction Section” with the sentence bellowed:

 “The current standard of care for HCC patients with PVTT/IVCTT is sorafenib, which is recommended as the only treatment option in EASL-EORTC practical guideline

We also emphasize the importance of sorafenib in the “Discussion Section” with the sentence bellowed:

Sorafenib has been considered as the first-line treatment for HCC patients with macrovascular invasion in most guidelines. Past studies suggest TACE-sorafenib combination therapy could be a promising treatment combination for advanced stage HCC”

We hope adding these sentences can better emphasize the importance of Sorafenib in the management of HCC patients with MVI.

Point 3. I'm not sure that radiological endpoints (PFS, response rates) are easy to assess when dealing with MVI and SBRT treatment. The authors state that they assessed response of MVI separately by mRECIST. Most of their patients had type I to II PVT, which would be most of the time not target lesion by mRECIST (less than 25px, difficult to ensure arterial enhancement). Moreover, in my experience, evaluation by mRECIST is particularly difficult following SBRT, with frequent inflammation mimicking arterial enhancement of the lesion. Comparison of PFS between a SBRT and a non-SBRT arm are particularly prone to evaluation bias. I would suggest to amend the text to recognize difficulty of evaluation as limitations, and please clarify if MVI was really evaluated by mRECIST as target lesions?

Response 3: We agree with your comments that it is somewhat difficult to evaluate the impact of SBRT treatment on tumor thrombosis using mRECIST criteria, although it is proposed by many previous studies. Currently there is no standard imaging protocol for evaluation of HCC treated with SBRT. According to mRECIST, Tumor Thrombosis (TT) should be considered a nontarget lesion. Complete response is defined as the disappearance of TT; incomplete response/stable disease is the persistence of TT; and progressive disease is the unequivocal progression of TT. According to your suggestions, we have clarified this point in the “Methods Section”, revised the associated results and conclusion in “Abstract” and have revised the evaluation results in Table 4 as shown in bellowed in the revised manuscript:

Table 4. Best treatment response at 24 weeks since first TACE in the two treatment groups after propensity score matching

CR (n)

PR (n)

SD (n)

PD (n)

NA

RR (%)

P*

Overall   Response

0.139

    TACE-SBRT

1

10

5

7

3

42.3

    TACE-sorafenib

1

5

6

8

5

23.1

Response   of Hepatic lesions

0.575

    TACE-SBRT

4

8

5

6

3

46.2

    TACE-sorafenib

2

8

3

7

5

38.5

Response   of Macrovascular Invasion

/

    TACE-SBRT

3

18a

2

3

/

    TACE-sorafenib

1

15a

4

5

/

Note: a Sum of number of cases with PR or SD. *P value was calculated by comparing the RR of two groups using Chi-square test.

Point 4. As the treatment was performed in one single center, there is a risk for selection bias. This has been mitigated by propensity score matching. However, could the authors better explain how the treatment modalities were proposed to patients?

Response 4: Many thanks for this question. In this study, each case was fully evaluated by a multidisciplinary team before initiation of treatment, including interventional radiologists, oncologists, pathologists, radiologists and radiation oncologists. After that, each patient will be informed of the recommended treatment plans, including sorafenib monotherapy, TACE-sorafenib therapy, TACE-SBRT therapy, et al. Every patient will be informed that sorafenib is the current first-line treatment recommended by guidelines and the expected survival benefits from taking it, as well as the potential side effects. For proposing TACE-sorafenib combination treatment, each patient will be informed that sorafenib will be started to taken immediately after the first TACE. For proposing the TACE-SBRT combination treatment, before first TACE, each patient will be informed that combined SBRT treatment will start immediately after 1-2 repeats of TACE. The SBRT is expected to bring a good local control of tumor thrombus but may also increase the risk of radiation-induced liver disease (RILD). After the patient was informed of the advantages and disadvantages of these treatment options, including treatment outcomes, treatment-related morbidities, and costs, the final treatment decision was made jointly by doctors and patients.

We have added some details in the description of treatment selection in the “Methods Section” of the revised manuscript as bellowed:

In this study, each case was fully evaluated by a multidisciplinary team before initiation of treatment, including interventional radiologists, oncologists, pathologists, radiologists and radiation oncologists. Each patient was informed of the advantages and disadvantages of the two treatment plans before initial TACE, including costs, treatment-related morbidities, and treatment outcomes. The decision of treatment plan was made jointly by the patient and doctors.

Minor comments:

Point 1. Please clarify the presence of metastasis as an exclusion criteria

Response 1: Many thanks for this suggestion. In our previous manuscript, the first exclusion criteria is “clinically or pathologically diagnosed local regional lymph node metastasis or distant metastasis”, now we changed into a more clarified wording as below:

“a) presence of extrahepatic metastasis detected on abdominal CT/MRI scan or chest CT scan/digital chest radiograph

Point 2. Please add in the introduction and/or discussion the median OS obtained for patients with MVI for sorafenib in the SHARP (about 8 months) and if available in the AP trial

Response 2: Many thanks for this kind advice. We found the mOS for patients with MVI after sorafenib monotherapy in the second analysis of SHARP trial but could not find the exact mOS for patients with MVI after sorafenib monotherapy in the AP trial. Therefore, we revised the first paragraph of “Introduction section” as below:

Macroscopic vascular invasion of hepatocellular carcinoma (HCC), including portal vein tumor thrombosis (PVTT) or inferior vena cava tumor thrombosis (IVCTT), corresponds to Barcelona Clinic Liver Cancer (BCLC) stage C disease and remains a daunting challenge for treatment [1,2]. The current standard of care for HCC patients with PVTT/IVCTT is sorafenib, which is recommended as the only treatment option in EASL-EORTC practical guideline. However, only modest clinical efficacy was observed when adopting sorafenib monotherapy for HCC patients with macrovascular invasion (MVI), with a median overall survival time of 8.1 months in the SHARP trial [4].

We hope this change can better clarify the efficacy of sorafenib in HCC with MVI.

Point 3. I was not able to review the suppl material table; however the figures are give in the text, so I'm not sure it adds any imprtant data.

Response 3: Below is the content of our supplemental table. Although the data is not that important, and we think it’s necessary to show it in the supplement of the paper.

Table S1. Number of repeats of TACE in each treatment group

Number of repeats

TACE-SBRT (n, %)

TACE-sorafenib (n, %)

Before SBRT

After SBRT

0

0 (0.0)

0 (0.0)

0 (0.0)

1

13 (50.0)

1 (3.8)

21 (43.8)

2

13 (50.0)

2 (7.7)

14 (29.2)

3

0 (0.0)

1 (3.8)

5 (10.4)

4

0 (0.0)

0 (0.0)

4 (8.3)

5

0 (0.0)

0 (0.0)

2 (4.2)

6

0 (0.0)

0 (0.0)

2 (4.2)

Point 4. Radioembolization is an alternative for such patients, and could be discussed.

Response 4: Many thanks for this kind suggestion. As the first reviewer also proposed to include the discussion of DEB-TACE versus cTACE in the manuscript, we added a paragraph in discussing both DEB-TACE and Radioembolization in the“Discussion Section”as bellowed:

In Asian regions, TACE is considered the primary treatment for patients with unresectable HCC confined in the liver. Two randomized trials comparing TACE to conservative treatment in HCC patients with PVTT had consistently demonstrated favorable OS in the TACE arm. Following efforts had been made to improve the efficacy of embolization for unresectable HCC through using drug-eluting beads and yttrium-90 radio-embolization, while the survival benefits were limited as reported by a recent published network meta-analysis.